# Prevalence and Social Determinants of Food Insecurity among College Students during the COVID-19 Pandemic

**DOI:** 10.3390/nu12092515

**Published:** 2020-08-20

**Authors:** Meghan R. Owens, Francilia Brito-Silva, Tracie Kirkland, Carolyn E. Moore, Kathleen E. Davis, Mindy A. Patterson, Derek C. Miketinas, Wesley J. Tucker

**Affiliations:** 1Department of Nutrition and Food Sciences, Texas Woman’s University, Houston, TX 77030, USA; mowens14@twu.edu (M.R.O.); cmoore8@twu.edu (C.E.M.); mpatterson14@twu.edu (M.A.P.); dmiketinas@twu.edu (D.C.M.); 2Department of Nutrition and Food Sciences, Texas Woman’s University, Denton, TX 76204, USA; fbritosilva@twu.edu (F.B.-S.); KDavis10@twu.edu (K.E.D.); 3Department of Nursing, University of Southern California, Los Angeles, CA 90089, USA; TKirkland4@twu.edu; 4Institute for Women’s Health, College of Health Sciences, Houston, TX 77030, USA

**Keywords:** food security, hunger, females, housing insecurity, unemployment, coronavirus

## Abstract

The coronavirus disease (COVID-19) pandemic has increased unemployment and food insecurity in the United States (US). Prior to the pandemic, college students exhibited higher rates of food insecurity than nonstudent households. The objectives of this study were to assess the prevalence and determinants of food insecurity among college students during the COVID-19 pandemic. We administered an online survey to 651 students on three diverse campuses at a state-funded university in Texas, US, in May 2020. Food security was assessed using a multistep approach that included the 2-item Food Sufficiency Screener and 6-Item USDA Food Security Survey Module (FSSM). Overall, 34.5% of respondents were classified as food insecure within the last 30 days. The strongest predictors of food insecurity were change in current living arrangement (OR = 2.70, 95% CI: 2.47, 2.95), being furloughed (OR = 3.22, 95% CI: 2.86, 3.64), laid off (OR = 4.07, 95% CI: 3.55, 4.66), or losing part-time work (OR = 5.73, 95% CI: 5.09, 6.46) due to the COVID-19 pandemic. These findings highlight the high prevalence of food insecurity among college students during the COVID-19 pandemic, with students who experienced housing insecurity and/or loss of income due to the pandemic being impacted the most.

## 1. Introduction

Food insecurity, defined as limited or uncertain access to sufficient, nutritious food for an active, healthy life [1], is associated with poor dietary quality [2,3,4] and numerous chronic diseases [5,6,7,8,9,10]. The coronavirus disease (COVID-19) pandemic has had a profound deleterious effect on health and economic wellbeing in the United States. In addition to the catastrophic loss of life associated with COVID-19 [11], local and state-wide mandates (e.g., stay-at-home orders and business closures) aimed at reducing virus spread have resulted in large increases in unemployment, food insecurity, and hunger [11,12,13,14]. Prior to the pandemic, 11.1% of the US households were classified as food insecure [15]. However, early reports from data collected during the COVID-19 crisis suggest that food insecurity has rapidly risen above prepandemic levels [12,14,16]. Indeed, survey data collected in late April 2020 indicate that the overall rates of the US household food insecurity have effectively doubled during the COVID-19 pandemic [12]. Furthermore, it appears that high-risk populations for food insecurity may be disproportionately affected by COVID-19 [14]. A national survey conducted in low-income adults (<250% federal poverty line) in the United States in mid-March 2020 revealed that 44% were food insecure and those individuals were more likely to be Black or Hispanic and have children in the home [14]. College students are another high-risk population for food insecurity who may be disproportionately impacted by the COVID-19 pandemic.

Studies consistently show that college students have higher rates of food insecurity than nonstudent US households [17,18,19,20,21,22,23,24,25,26,27]. Food-insecure college students are more likely to be younger [22,24,28], Black or Hispanic [18,21,24,27,29,30], low-income [29,31], employed [29,31], receiving financial aid [27,32], and housing insecure [27]. Food insecurity has also been associated with lower dietary quality [18,33], poor mental health [19,21], and lower academic performance [18,21,34] among students. There are several unique features that place college students at an elevated risk for food insecurity during the COVID-19 crisis (relative to the US population). First, many college students work part-time or full-time in the service industry (restaurants, bars, and healthcare), which has been one of the hardest hit economic sectors of the US economy during the COVID-19 crisis [35]. Second, many college students do not qualify for federal food assistance programs that help to combat food insecurity due to loss of employment such as the Supplemental Nutrition Assistance Program (SNAP; formerly known as food stamps). SNAP requires that applicants work 20 h per week for three or more months within the last 36 weeks prior to the application [36]. This requirement often disqualifies many college students who are unable to work 20 h or more per week while attending university. Furthermore, participation in SNAP is relatively low among college students who do qualify due to lack of awareness regarding eligibility, increased administrative burden, and the social stigma associated with accepting welfare [37]. Third, while many Americans received federal stimulus checks of up to $1200 plus $500 for each child as part of the $2 trillion COVID-19 relief package called the Coronavirus Aid, Relief, and Economic Security (CARES) Act [38], many college students were ineligible to receive a stimulus payment of their own because they were claimed as dependents on their parents’ tax returns. Finally, many college students possess low food literacy, practically defined as the ability to plan, shop for, prepare, and cook nutritionally balanced meals [39,40]. COVID-19 forced the closure of campus dining halls and cafeterias, leaving many college students to purchase and prepare meals on their own. Taken together, this unique set of factors has placed college students at a high risk for food insecurity during the COVID-19 pandemic in the United States.

To date, no study has explored the impact of COVID-19 on food insecurity in college students. Therefore, the primary objective of this study was to assess the prevalence of food insecurity among college students at a university in Texas, US, during the COVID-19 pandemic. A secondary objective was to examine the relationships between food insecurity, sociodemographic, and economic factors, including change in employment and living status as a direct result of the COVID-19 pandemic.

## 2. Materials and Methods

### 2.1. Study Design and Participants

This study was cross-sectional and used an online survey. We designed a web-based survey using the online survey tool in PsychData (PsychData LLC, State College, PA, USA) to administer surveys to college students to assess the prevalence of food insecurity during the COVID-19 pandemic. The online survey was conducted at all three campuses of a large, diverse, state-funded university in Texas, US. Two of the campuses are located in large metropolitan areas (Houston and Dallas), and the third campus is located in a suburban area (Denton) 40 miles northwest of Dallas, TX. In 2020, this university had a total enrollment of 14,888 students across three campuses (88% female, 38% graduate, 62% undergraduate, 40% White, 27% Hispanic, 18% Black, and 11% Asian/Pacific Islander). All current undergraduate and graduate students at all three campuses were eligible to participate in the study if they were 18 years or older, were fluent in English, and had access to the internet. Survey responses were solicited from all students by sending a weekly email containing a study flyer with a hyperlinked URL to our survey using the university student listserv. The survey was open to participants from 12 May to 12 June 2020. A timeline of the state of Texas response to the COVID-19 pandemic and our survey administration is provided in Figure 1**.** All study procedures were approved by the institutional review board for human subjects research at Texas Woman’s University (Houston, TX, USA) (IRB#: FY2020-314). After reviewing a cover page that included a brief description of the study purpose, estimated time to complete the survey, and inclusion criteria for participation, respondents agreed to participate in the study by selecting “When you click continue below, the return of your completed survey or completing the online survey constitutes your informed consent to act as a participant in this research.” The online survey was designed to be completed in approximately 15 min on a personal computer or mobile device. All participants who completed the survey were entered into a drawing to win one of 25 $50 digital gift cards.

### 2.2. Food Security and Sociodemographic Characteristics Survey

Food security in the last 30 days was assessed using the validated 2-item Food Sufficiency Screener [44] and 6-item USDA Food Security Survey Module (FSSM) [45]. Although the 6-item USDA FSSM has been used extensively by itself to assess food security in college students, this device is validated for use in the US households, and the validity of this instrument within college students is still unclear [26,46]. In a recent study among college students, the use of this multistep food security assessment approach produced estimates of food insecurity that were more consistent with known incomes compared to the 6-item USDA FSSM alone [26]. Furthermore, estimates of food insecurity among college students were almost 2-fold lower when assessed with the multistep food security assessment approach vs. the 6-item USDA FSSM alone [26]. All respondents were first presented with the 2-item Food Sufficiency Screener and if they responded with “No” and “Enough of the kinds of food we want to eat,” respectively, they were considered food secure and did not complete the 6-item USDA FSSM (i.e., screened out) (Table 1). Respondents who did not pass the 2-item Food Sufficiency Screener proceeded to complete the 6-item USDA FSSM. The sum of affirmative responses to six questions on the 6-Item USDA FSSM was used to calculate a raw score that determined the level of food security using the USDA coding scheme [45] as follows: food secure (score 0–1), low food security (score 2–4), or very low food security (score 5–6).

Following completion of the 2-item Food Sufficiency Screener and 6-item USDA FSSM (for those who did not pass the 2-item screener), a variety of sociodemographic characteristics were collected including a participant’s age, height and weight, sex, race/ethnicity, college undergraduate or graduate status, campus location attended or online only, and current participation in either SNAP, Women, Infant, and Children (WIC), or another type of food assistance program. The final four questions of the survey were aimed at better understanding some of the socioeconomic factors that may have directly impacted food security during the COVID-19 pandemic:What is your current living arrangement?Has coronavirus (COVID-19) directly impacted your current living arrangement?Has coronavirus (COVID-19) directly impacted your current employment status?If you answered “Yes” to the prior question, how has your income been affected?oFurloughedoLaid offoLost shift workoNot applicable, my income is the sameoOther (please specify).

### 2.3. Data Analysis

Data were poststratified to reflect the sex and race/ethnicity demographics of our institution in Texas, US. Sample weights were generated by first calculating a base weight equal to the university student population divided by the sample size (N/n). Next, the sum of the provisional sample weights was calculated for each sex by race/ethnicity combination for the sample and divided by the counts for each sex by race/ethnicity combination for the population. This yielded poststratification factors for each sex by race/ethnicity combination. The final weights were calculated by multiplying each respondents’ base weight by their respective sex by race/ethnicity poststratification factors. We provide both weighted and unweighted estimates for all analyses. Differences in categorical variables were analyzed using Chi-squared tests for nominal variables and Cochran–Mantel–Haenszel test for ordinal variables. LASSO logistic regression was used to identify predictors of food insecurity for both weighted and unweighted data. Food insecurity was treated as a binary variable for the logistic regression with the following levels: food insecure (USDA FSSM response > 1) and food secure (USDA FSSM response ≤ 1). Data were grouped over 7-day intervals to examine the change in food security prevalence and COVID-19 pandemic impact on employment over time. All analyses were conducted using SAS software version 9.4 (SAS Institute Inc., Cary, NC, USA).

## 3. Results

### 3.1. Participant Characteristics

Among the 14,888 students eligible for participation across the three campuses, a total of 651 responded to the online survey (overall response rate = 4.4%). Of these, 137 respondents did not complete all survey questions and 12 completed the survey twice (the second response was removed) resulting in a final sample of 502 participants. Sociodemographic and other characteristics of survey respondents are presented in Table 2.

Survey respondents were primarily female (93.6%), white (50.4%), and enrolled in classes on the Denton campus (50.2%), which is similar to the campus population overall (87.5% female, 40.3% white, and 83.5% of total enrollment in Denton) (Table 2). Most respondents were graduate students (56.5%), which contrasts with the overall campus population (37.8% graduate students). The majority of survey respondents (60.4%) were classified as normal weight (BMI < 25 kg/m^2^) according to BMI criteria calculated using self-reported height and weight. Participation in food assistance programs was low, with 2.9% and 1.6% of respondents reporting current participation in SNAP and WIC, respectively.

### 3.2. Food Security during the COVID-19 Pandemic

The unweighted and weighted prevalence of food security among survey respondents, measured using a multistep food security assessment approach, is presented in Table 3. Overall, 65.5% of students failed to pass the 2-item Food Sufficiency Screener and were at-risk for food insecurity. Among these students, low (30.8%) and very low (21.9%) food security was highly prevalent according to the 6-item USDA FSSM. To assess food security for our entire sample, we added back the respondents who were classified as food secure by passing the 2-item Food Sufficiency Screener (34.5% of students) to those already classified according to the 6-item USDA FSSM. For the overall sample, 34.5% of students were considered food insecure (20.2% low food secure and 14.3% very low food secure).

Frequency of responses to individual questions on the 2-item Food Sufficiency Screener and 6-item USDA FSSM are provided as weighted and unweighted data in Appendix A. Of note, in those who responded to the 6-item USDA FSSM, 38.2% reported reducing the size of meals or skipping meals because there was not enough money for food and 26.7% reported doing this on 3 or more days during the last 30 days. Furthermore, 46.7% of respondents reported eating less than they felt they should because there was not enough money for food and 20.4% reported experiencing hunger and not eating because there was not enough money for food in the last 30 days. Finally, 55.1% of students responded “often true or sometimes true” to the question “I/we could not afford to eat balanced meals in the last 30 days.”

### 3.3. Sociodemographic Correlates of Food Insecurity during the COVID-19 Pandemic

Significant associations were noted when comparing food-insecure and food-secure students on sociodemographic characteristics and whether or not their living arrangement or employment was directly impacted by the COVID-19 pandemic (Table 4). Accordingly, undergraduate students, minority students (including Black, Hispanic, and Asian/Pacific Islander), single parents, and students enrolled in classes on the Denton campus were more likely to be food insecure (*p* < 0.001). In addition, food-insecure students were more likely to be younger (*p* = 0.02) and overweight or obese (*p* < 0.001).

Approximately 1 in 4 students reported having their current living arrangement directly impacted by the COVID-19 pandemic, and these students were more likely to be food insecure (*p* < 0.001) (Table 4). Over half of students reported that their current employment status had been directly impacted by the COVID-19 pandemic, and these students were also more likely to be food insecure (*p* < 0.001). For those students who reported their current employment status impacted by the COVID-19 pandemic, 13.9% were furloughed, 9.9% were laid off, 13.4% lost part-time shift work, and 18.5% listed “other” as an impact on income. The three most common responses for listing “other” for “how has income been affected?” were: (1) reduced work hours, (2) resigned or quit job due to unsafe working conditions and/or presence of a pre-existing condition, and (3) spouse or parent lost their job or were furloughed. Students who reported a negative impact on income (furloughed, laid off, lost part-time shift work, other) as a direct result of COVID-19 were more likely to be food insecure (*p* < 0.001).

### 3.4. Predictors of Food Insecurity during the COVID-19 Pandemic

A LASSO logistic regression was conducted to identify possible predictors of food insecurity based on BMI, age, race/ethnicity, class status, current living arrangement, and changes in current living arrangement and/or employment status as a result of the COVID-19 pandemic (Table 5). Compared to White students, Black (OR = 1.61, 95% CI: 1.45, 1.80), Hispanic (OR = 1.90, 95% CI: 1.72, 2.09), and Asian/Pacific Islander (OR = 2.12, 95% CI: 1.86, 2.42) students had greater odds of being food insecure (*p* < 0.001, all comparisons). Undergraduate students had greater odds for food insecurity (OR = 1.20, 95% CI: 1.11, 1.31) compared to graduate students (*p* < 0.001). Students who were overweight/obese had greater odds of being food insecure (OR = 1.45, 95% CI: 1.34, 1.57) compared to normal-weight students (*p* < 0.001). Compared to students who lived alone, students who were single parents had greater odds of being food insecure (OR = 2.48, 95% CI: 2.03, 3.04) (*p* < 0.001). In contrast, students who were currently living with parents or other relatives had lower odds of being food insecure (OR = 0.50, 95% CI: 0.43, 0.58) than students who were living alone (*p* < 0.001).

Students who reported that their current living arrangement was directly impacted by the COVID-19 pandemic had 2.70 (95% CI: 2.47, 2.95) times greater odds of being food insecure compared to those who were not affected (*p* < 0.001). The strongest predictor of food insecurity was change in current employment status as a direct result of the COVID-19 pandemic (Table 5). Students who reported being furloughed, laid off, lost part-time shift work, or other changes in income as a direct result of the COVID-19 pandemic had 3.22 (95% CI: 2.86, 3.64), 4.07 (95% CI: 3.55, 4.66), 5.73 (95% CI: 5.09, 6.46), and 3.34 (95% CI: 3.01, 3.72) times greater odds of food insecurity compared to those whose income was not affected, respectively (*p* < 0.001, all comparisons).

### 3.5. Change in Food Insecurity Prevalence during the Data Collection Period

Given the predictive power of change in employment/income on food insecurity during the COVID-19 pandemic, we assessed whether the prevalence of food security changed over the duration of the data collection period. As shown in Figure 2, the prevalence of food insecurity decreased from 37.0% during week 1 (12 May–18 May) to 33.2% during week 4 (2 June–10 June) of data collection (*p* < 0.001). This coincided with more people reporting no change in income as a result of COVID-19 (40.6%) during week 4 compared to week 1 (37.3%; *p* < 0.001) of data collection.

## 4. Discussion

This study sought to assess the prevalence of food insecurity among college students at a university in Texas, US, during the COVID-19 pandemic. Using a multistep food security assessment approach, we found that over 1 in 3 college students experienced food insecurity during the last 30 days. This rate of food insecurity is ~15% higher than the rate of food insecurity previously reported at another US institution in 2019 using the same multistep food security assessment protocol in college students [26]. The high rates of food insecurity reported in the current study appear to be driven by change in living arrangement and/or loss of employment as a direct result of the COVID-19 pandemic. Specifically, 53.5% college students reported that their employment was directly impacted by the COVID-19 pandemic, and these students had 3.22 (furloughed) to 5.73 (lost part-time shift work) times greater odds of being food insecure than students whose income was not affected by the pandemic. Although food insecurity has increased to a similar extent in nonstudent US households [12,14] during the COVID-19 pandemic, it is worth noting that many college students do not qualify for many federal and state safety net programs (expanded SNAP benefits and federal stimulus). As a result, college students may be disproportionately impacted by food insecurity and without immediate assistance during the COVID-19 pandemic.

Prior to the COVID-19 pandemic, studies consistently showed that college students have higher rates of food insecurity than nonstudent US households [18,19,20,21,22,23,24,25,26,27,46,47]. In the current study, we found levels of food insecurity (34.5%) higher than those reported in the US households (~22%) during the COVID-19 pandemic [12]. In contrast, our rates of food insecurity are lower than those reported in college students in several studies published prior to the pandemic [24,31,32,48]. In fact, a 2019 systematic review reported an average food insecurity prevalence of 43.5% among college students [46]. However, the assessment of food insecurity prevalence in college students has typically not included the use of a screener, with the majority of studies using a 6-item USDA FSSM or 10-item USDA FSSM by itself to assess food insecurity [26,46,49]. This is significant because food insecurity estimates that use the 6-item USDA FSSM or 10-item USDA FSSM alone may overestimate the prevalence of food insecurity in college students [25,26]. In a recent study by Nikolaus and colleagues [26], the prevalence of food insecurity in college students at a large Midwestern US university was estimated to be 34.9% when assessed using the 6-item USDA FSSM alone. However, when the researchers used a multistep food insecurity assessment protocol that included the 2-item Food Sufficiency Screener to screen out food-secure students prior to administering the 6-item USDA FSSM, the food insecurity prevalence was only 19.9%. Although the rates of food insecurity reported in the current study are ~15% higher than those reported by Nikolaus and colleagues [26] using the same methodology, our sample of students included more females, minorities, and lower SES students making direct comparisons difficult. Given the limitations associated with comparisons to prior work and the nature of our study design, it is unclear if food insecurity in college students increased during the COVID-19 pandemic. However, given the profound impact of record unemployment on food insecurity rates in the United States overall [12,14] and higher rates of food insecurity reported by students who lost employment in the current study, we believe that food insecurity may have increased among college students during the COVID-19 pandemic. To better isolate the impact of the COVID-19 pandemic on food insecurity prevalence in college students, we will follow-up with the same subjects in 1 year to reassess food insecurity prevalence. However, the validity of this follow-up assessment will be contingent upon a return to a “normal” economic and social environment in the United States.

Using the 6-item USDA FSSM, food insecurity is further categorized into either low food security or very low food security [45]. Low food security is indicated by reduced quality, variety, or desirability of food in the diet with little or no reduced food intake. In contrast, very low food security is associated with disrupted eating patterns and reduced food intake. In 2018, 11.1% of the US households were food insecure with 6.8% and 4.3% of households being classified as low food secure and very low food secure, respectively [15]. In the current study, we found that 20.2% of students surveyed were low food secure and 14.3% were very low secure during the COVID-19 pandemic. Although high rates of low and very low food security are both concerning for overall health; the rate of very low food security is perhaps of greater concern given its association with hunger and reduced food intake. Indeed, when we looked specifically at the 6-item USDA FSSM questions that assessed hunger and reduced food intake, we found that 46.3% of respondents reported eating less than they felt they should because there was not enough money for food, and 20.8% of respondents reported experiencing hunger and not eating because there was not enough money for food in the last 30 days. Taken together, these findings suggest that the prevalence of hunger is high among college students during the COVID-19 pandemic.

Prior to the COVID-19 pandemic, food-insecure college students were more likely to be younger [22,24,28,30], Black or Hispanic [18,24,27,29,30,50], low-income [29,31], employed [29,31], receiving financial aid [27,32], and housing insecure [27]. In the current study, we observed similar relationships between the aforementioned sociodemographic characteristics and the likelihood of being food insecure. Specifically, younger, undergraduate, and minority students (including Black, Hispanic, and Asian/Pacific Islander) were more likely to be food insecure. Furthermore, when compared to White students, minority students had ~2 times greater odds of being food insecure. These findings are consistent with recently published US Census Bureau data, which show that Black and Hispanic households with children are now twice as likely to be food insecure when compared to White families during the COVID-19 pandemic [16]. Unfortunately, these racial disparities do not only pertain to differences in food insecurity, as Blacks and Hispanics are also at an increased risk for getting COVID-19 and experiencing adverse outcomes (hospitalization and death) when compared to White individuals [51]. We also found that single parent students were more likely to be food insecure and had over 2 times greater odds of food insecurity compared to students who lived alone. In contrast, students who lived with their parents or other relatives had half the odds of experiencing food insecurity compared to students who lived alone. These findings suggest that familial or spousal support (in terms of living arrangement) may have provided college students with greater protection against food insecurity during the COVID-19 pandemic. Family support increases food purchasing power and may improve the likelihood of college students regularly consuming balanced meals, as many college students have low nutrition literacy and poor cooking skills [24,39]. Indeed, it has previously been shown that college students who are financially independent and not receiving any form of financial support from family members are more likely to be food insecure [52]. As such, the loss of employment during the COVID-19 pandemic may have conceivably had a greater impact on food insecurity in financially independent college students.

In the current study, we found that the two strongest predictors of food insecurity in college students were change in current living arrangement and/or loss of employment as a direct result of the COVID-19 pandemic. Almost 1 in 4 students reported having their current living arrangement directly impacted by the COVID-19 pandemic and these students had 2.70 greater odds of being food insecure compared to students who did not have their living arrangement impacted. These findings are consistent with prior work in college students, which demonstrate that students who experience housing problems are significantly more likely to be food insecure [27]. Regarding the impact of loss of employment on food insecurity, we found that over half of the students surveyed had their current employment directly impacted by the COVID-19 pandemic and these students had roughly three to six times greater odds of being food insecure depending on whether they were furloughed, laid off, or lost part-time shift work. Although the impact of the COVID-19 pandemic on unemployment and food insecurity is not exclusive to just college students [14,16,35], the magnitude of unemployment and its predictive power on food insecurity is profound. In the US population, unemployment increased from 3.5% in February 2020 to 11.1% in 2020 June with food insecurity rising to ~22% [12,35]. This rise in unemployment and food insecurity was met with increased unemployment benefits, expansion of food assistance programs (SNAP), and federal stimulus checks. However, many college students did not qualify for these emergency assistance programs that brought food and financial assistance to struggling US households during the pandemic. Indeed, despite more than 1 in 3 students being food insecure in the present study, less than 5% of students reported current participation in food assistance programs (SNAP and WIC). SNAP is a major safety net to reduce food insecurity and improve intake of healthful foods [36]. Although many college students do not meet the criteria to participate in SNAP, a 2018 US Government Accountability Office report found that 57% of potentially eligible low-income students with food insecurity risk factors do not participate in the program [37]. This suggests that both expanded access to and increased awareness of SNAP may be needed to increase college student participation. One of the successful initiatives enacted specifically to combat food insecurity and hunger among college students in the United States is college food pantries [50]. However, many college food pantries (including the food pantry at our institution) were forced to close during the COVID-19 pandemic leaving students to search for food assistance elsewhere. Finally, while many Americans received federal stimulus checks of up to $1200 plus $500 each for a child as part of the $2 trillion COVID-19 relief package called the CARES Act [34], many college students were ineligible to receive a stimulus payment of their own as they were claimed as dependents on their parents’ tax returns. College students do not only have high rates of food insecurity but reduced access to the food and financial assistance programs that were created or expanded upon to combat food insecurity during the COVID-19 pandemic in the United States. As such, federal mechanisms specifically targeting food insecurity in college students are urgently needed.

In the United States, food insecurity is associated with an increased risk of obesity [7,53,54,55,56], type 2 diabetes [5,9,57], and metabolic syndrome [58]. In particular, women who are food insecure have been shown to have a greater likelihood of being overweight or obese relative to their food secure peers [55,56,59]. This association between food insecurity and obesity has come to be known as the “hunger-obesity paradox.” In contrast to the perception that inadequate access to food would cause a reduction in BMI, the hunger-obesity paradox posits that food insecurity may cause weight gain as a result of increased consumption of calorie-dense, nutrient-poor foods due to episodic undereating [60,61]. Although the number of studies is limited, food insecurity has also been shown to be associated with overweight/obesity in college students [23,33]. Furthermore, a recent study in 547 university students in North Carolina demonstrated that students with lower food security were more likely to engage in potentially obesogenic coping behaviors such as purchasing and consuming cheap, processed foods and overeating when food was plentiful [62]. In the current study, we found that food-insecure college students were on average 2 BMI units heavier than food-secure students. Furthermore, students who were overweight/obese had greater odds of being food insecure compared to normal-weight students during the COVID-19 pandemic. It is unclear if a higher prevalence of food insecurity among students who were overweight/obese resulted in worse dietary intake quality during the COVID-19 pandemic. As such, future research is needed to explore whether increases in food insecurity are associated with adverse changes in overall diet quality and whether this differs by BMI, during a pandemic.

### Strengths and Limitations

This study has several strengths. First, the use of a multistep food security assessment protocol may provide greater relative accuracy for identifying students with food insecurity as opposed to using the 6-item or 10-item USDA FSSM alone, which may have the tendency to overestimate the prevalence of food insecurity in college students [26]. Second, our survey sample was weighted to ensure that it was representative of the campus population for sex and race/ethnicity. Third, our survey sample was racially diverse and included students from all three campuses located in Texas’s two largest metroplexes (Dallas-Fort Worth and Houston). This is important because Black and Hispanic college students have been shown to be at an increased risk for food insecurity relative to White students [18,21,24,27,29,30]. Finally, the addition of questions related to changes in living arrangement and/or changes in income as a direct result of the COVID-19 pandemic provided better insight as to what the main driving factors behind food insecurity were during this pandemic. Although certain sociodemographic factors (race/ethnicity, BMI, class status, and campus location) were predictive of food insecurity, change in living arrangement and/or loss of income were the strongest predictors of food insecurity during the COVID-19 pandemic.

This study has several limitations. First, this survey was cross-sectional and as such does not provide causation that the prevalence of food insecurity observed in our sample is exclusively the result of the COVID-19 pandemic. However, given the predictive power of change in current living arrangement and/or loss of employment on food insecurity in our survey sample, we believe that the COVID-19 pandemic is a likely contributor to the high rates of observed food insecurity. Second, our survey response rates were low. It is unclear why our response rates were low; however, similarly low response rates have been reported by other investigators [25,31,63] assessing food insecurity prevalence in college students. A potential contributor to the low response rates in the current study could be the timing of our survey. The survey request was emailed to students at the beginning of the summer (mid-May) when many students were not actively taking courses or checking their university email. Despite lower response rates, the students who responded to the survey were representative of the overall student body in terms of sex and race/ethnicity. Third, survey respondents could choose whether or not to participate in the survey, which may have introduced selection bias. We tried to mitigate selection bias by encouraging all students to participate in our study regardless of whether or not they had noticed any differences in dietary behavior during the pandemic. In addition, all students were notified prior to participation that completion of the survey would enter them into a drawing to win a $50 gift card. Fourth, there are other determinants of food security that were not assessed in this study including nutrition and food literacy [24,39]. It is plausible that poor cooking skills, food shortages, and fear of grocery shopping may have contributed to food insecurity among college students. As such, future studies are needed to explore how the COVID-19 pandemic impacted food accessibility, food selection, food preparation, and overall dietary quality in college students. Finally, these findings may not be generalizable to college students across the United States as our sample is limited to one primarily female, diverse, state-funded university located in Texas, US. It is unclear if food insecurity prevalence would be similar at other US universities that have equal representation by sex and a race/ethnicity makeup that more closely approximates the US demographics overall.

## 5. Conclusions

In conclusion, this study found a high prevalence of food insecurity among college students surveyed at a university in the United States during the COVID-19 pandemic. The two strongest predictors of food insecurity in college students were change in living arrangement and/or loss of employment as a direct result of the COVID-19 pandemic. A robust, comprehensive policy response is needed to mitigate food insecurity in college students during the COVID-19 pandemic in the United States.

## Figures and Tables

**Figure 1 nutrients-12-02515-f001:**
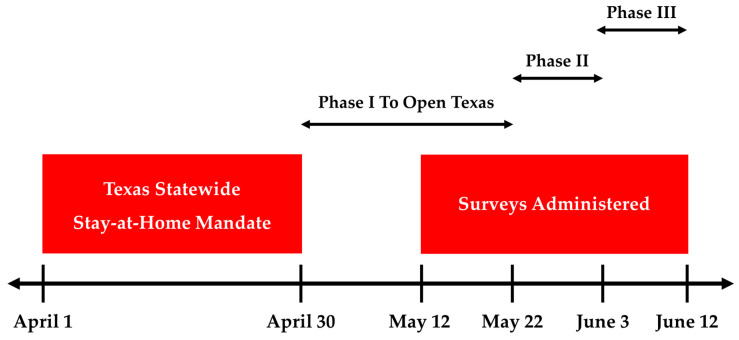
Survey data collection timeline and Texas statewide mandates in response to COVID-19, and phased openings. Phase I: restaurants permitted to open at 25% capacity [41]. Phase II: restaurants permitted to open at 50% capacity. Bars and bowling alleys permitted to open at 25% capacity. Gyms, offices, and manufacturing businesses permitted to open at 25% capacity on 18 May [42]. Phase III: all businesses permitted to open at 50% capacity. Restaurants permitted to open at 75% capacity [43].

**Figure 2 nutrients-12-02515-f002:**
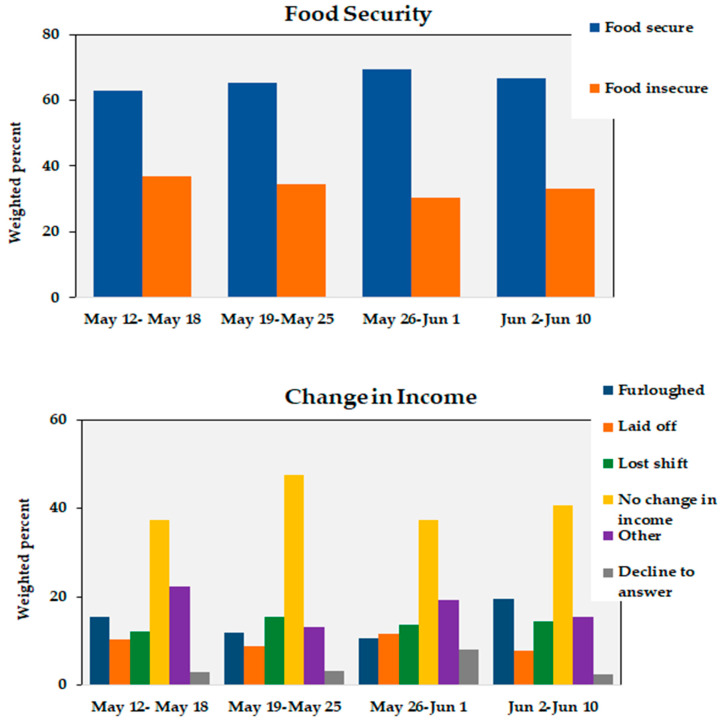
Food security prevalence (top) and change in income (if employment affected) (bottom) as a direct result of the COVID-19 pandemic during the data collection period.

**Table 1 nutrients-12-02515-t001:** Food security survey questions and response coding.

Item/Questions	Affirmative Responses (Food Insecure)	Negative Responses (Food Secure)
**2-Item Food Sufficiency Screener**		
In the last 30 days, did you ever run short of money and try to make your food, or your food money go further?	Yes	No
In the last 30 days, which of these statements best describes the food eaten in your household?	Enough but not always the kinds of food we want to eat, sometimes not enough to eat, often not enough to eat	Enough of the kinds of food we want to eat
If participants responded negatively to both questions above (2-item Food Sufficiency Screener) they were screened out and did not complete the 6-item USDA Food Security Survey Module.		
**6-Item USDA Food Security Survey Module: Short Form**		
In the last 30 days, did you/other adults in your household ever reduce the size of your meals or skip meals because there was not enough money for food?	Yes	No
In the last 30 days, how often did you/other adults in your household reduce the size of your meals or skip meals because there was not enough money for food?	≥ 3 days	< 3 days
In the last 30 days, did you ever eat less than you felt you should because there was not enough money for food?	Yes	No
In the last 30 days, did you experience hunger and did not eat because there was not enough money for food?	Yes	No
In the last 30 days, the food that I/we bought just did not last, and I/we did not have enough money to get more.	Often true, sometimes true	Never true, do not know
In the last 30 days, I/we could not afford to eat balanced meals.	Often true, sometimes true	Never true, do not know

Sources: The US Food Security Survey Module: Six-Item Short Form [45], 2-Item Food Sufficiency Screener in Hager et al. [44], and Nikolaus et al. [26].

**Table 2 nutrients-12-02515-t002:** Sociodemographic and other characteristics of survey respondents (*n* = 502).

Descriptive Variables	Survey Unweighted		Survey Weighted	Total Campus
	*n*	%	%	%
**Sex**				
Female	466	93.6	87.5	87.5
Male	32	6.4	12.5	12.5
Total	498	100	100	100
**Race/ethnicity**				
White	247	50.4	40.4	40.3
Black	50	10.2	18.0	18.0
Hispanic	93	19.0	26.9	27.1
Asian/Pacific Islander	67	13.7	10.3	10.6
Other	33	6.7	4.4	4.0
Total	490	100	100	100
**Campus**				
Denton	252	50.2	52.1	83.5
Dallas	64	12.8	11.3	8.5
Houston	70	13.9	15.1	8.0
Online	107	21.3	20.2	N/A
Decline	9	1.8	1.3	N/A
Total	502	100	100	100
**Class status**				
Graduate	283	56.5	54.6	37.8
Undergraduate	216	43.1	44.9	62.1
Decline	2	0.4	0.4	N/A
Total	501	100	100	100
**Body Mass Index (kg/m^2^)**				
Normal weight (<25 kg/m^2^)	303	60.4	58.5	N/A
Overweight/obese (≥25 kg/m^2^)	199	39.6	41.5	N/A
Total	502	100	100	N/A

Note: Totals not adding up to total sample size are due to missing data/declined responses. Weighted values are weighted for sex and race/ethnicity. N/A: not available.

**Table 3 nutrients-12-02515-t003:** Food security among survey respondents measured using a multistep food security assessment approach that included the 2-item Food Sufficiency Screener and 6-item USDA Food Security Survey Module (FSSM).

Survey Instruments and Outcome Variables	Survey Unweighted		Survey Weighted
	*n*	%	%
**2-Item Food Sufficiency Screener (*n* = 502)**			
Passed	183	36.4	34.5
Failed	319	63.6	65.5
**6-Item USDA Food Security Survey Module (*n* = 319)**			
High/marginal food security	155	48.6	47.3
Low food security	96	30.1	30.8
Very low food security	68	21.3	21.9
**Overall Sample Food Security (2-item + 6-item) (*n* = 502)**			
High/marginal food security	338	67.3	65.5
Low food security	96	19.1	20.2
Very low food security	68	13.6	14.3

Note: Weighted values are weighted for sex and race/ethnicity.

**Table 4 nutrients-12-02515-t004:** Food security status (secure or insecure) according to sociodemographic characteristics, BMI, change in living status, and change in employment caused by the COVID-19 pandemic.

Sociodemographic and Other Characteristics	Total (*n* = 502)	Food Secure (*n* = 338)	Food Insecure (*n* = 164)	*p*-Value *
Sex				0.001
Female	87.5	88.1%	86.3%	
Male	12.5%	11.9%	13.7%	
**Race/ethnicity**				<0.001
White	40.4%	45.0%	31.6%	
Black	18.0%	16.3%	21%	
Hispanic	26.9%	24.6%	31.4%	
Asian/Pacific Islander	10.3%	9.5%	11.9%	
Other	4.4%	4.6%	4.1%	
**Campus**				<0.001
Denton	52.1%	48.4%	59.2%	
Dallas	11.3%	11.4%	11.2%	
Houston	15.1%	16.1%	13.1%	
Online	20.2%	22.8%	15.3%	
**Class status**				<0.001
Graduate	54.6%	59.7%	45.0%	
Undergraduate	44.9%	39.8%	54.6%	
**Age (years)**				0.02
Mean ± SEM	27.5 ± 0.4	28.2 ± 0.5	26.3 ± 0.6	
**BMI (kg/m^2^)**				0.006
Mean ± SEM	25.8 ± 0.3	25.1 ± 0.3	27.1 ± 0.7	
BMI classification				<0.001
Normal weight (BMI < 25.0) (%)	58.5%	61.8%	52.1%	
Overweight/obese (BMI ≥ 25.0) (%)	41.5%	38.2%	47.9%	
**Current living arrangement**				<0.001
Live alone	9.5%	9.9%	8.8%	
Live with roommates and/or spouse	44.1%	44.1%	44.2%	
Live with parents or other relatives	40.7%	41.7%	39.5%	
Live with my children only	5.3%	4.2%	7.5%	
**Has COVID-19 pandemic directly impacted current living arrangement?**				<0.001
Yes	23.5%	17.2%	35.6%	
No	75.7%	82.4%	62.9%	
**Has COVID-19 pandemic directly impacted current employment status?**				<0.001
Yes	53.5%	44.3%	70.9%	
No	46.4%	55.5%	29.1%	
**How has income been affected?**				<0.001
Not applicable, income unchanged	40.5%	50.6%	21.3%	
Furloughed	13.9%	12.7%	16.2%	
Laid off	9.9%	7.4%	14.5%	
Lost part-time shift work	13.4%	9.1%	21.5%	
Other (specify) ^a^	18.5%	16.3%	22.6%	

Notes: Frequencies not adding up to 100 due to missing data/declined responses. Data presented as weighted values weighted for sex and race/ethnicity. * *p*-values represent frequency differences between food secure and food insecure. ^a^ Other included: (1) reduced work hours, (2) resigned or quit job due to unsafe working conditions and/or presence of a pre-existing condition, and (3) spouse or parent lost their job or were furloughed. BMI: Body Mass Index, SEM: standard error of mean.

**Table 5 nutrients-12-02515-t005:** Predictors of food insecurity during the COVID-19 pandemic.

Effect	Odds Ratio Estimate (95% CI)	*p*-Value
**Race/ethnicity**		
Black vs. White	1.61 (1.45, 1.80)	<0.001
Hispanic vs. White	1.90 (1.72, 2.09)	<0.001
Asian/Pacific Islander vs. White	2.12 (1.86, 2.42)	<0.001
Other race/ethnicity vs. White	1.34 (1.11, 1.63)	0.003
**Class status**		
Undergraduate vs. graduate student	1.20 (1.11, 1.31)	<0.001
**Age (years)**	0.97 (0.96, 0.98)	<0.001
**BMI (kg/m^2^)**		
Overweight/obese vs. normal weight	1.45 (1.34, 1.57)	<0.001
**Current living arrangement**		
Live with roommates and/or spouse vs. I live alone	0.97 (0.84, 1.11)	0.61
Live with parents, or other relatives vs. I live alone	0.50 (0.43, 0.58)	<0.001
Live with my children only vs. I live alone	2.48 (2.03, 3.04)	<0.001
COVID-19 pandemic impacted current living arrangement? Yes vs. no	2.70 (2.47, 2.95)	<0.001
**Change in current employment status due to COVID-19 pandemic**		
Furloughed vs. income/employment unchanged	3.22 (2.86, 3.64)	<0.001
Laid off vs. income/employment unchanged	4.07 (3.55, 4.66)	<0.001
Lost part-time shift work vs. income/employment unchanged	5.73 (5.09, 6.46)	<0.001
Other changes in income ^a^ vs. income/employment unchanged	3.34 (3.01, 3.72)	<0.001

Notes: Data presented as weighted values weighted for sex and race/ethnicity. ^a^ Other included: (1) reduced work hours, (2) resigned or quit job due to unsafe working conditions and/or presence of a pre-existing condition, and (3) spouse or parent lost their job or were furloughed. BMI: Body Mass Index.

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
