# Peer review of "Prevalence and Social Determinants of Food Insecurity among College Students during the COVID-19 Pandemic"

_nutrients, 2020, doi:10.3390/nu12092515_

Round 1

Reviewer 1 Report

The first aim of the paper is to assess the prevalence of food insecurity among college students in a university in Texas (US) during the COVID-19 pandemic, while the secondary goal – that I find even more interesting - was to examine the relationships between food insecurity, sociodemographic, and economic drivers, including changes due the COVID-19 pandemic.. A Logistic Regression has been estimated on a sample of 502 students.

The paper is very interesting and well written.

Some aspects need to be better clarified. More in detail:

Introduction:

  • Lines 31: Actually, the reported definition refers to the World Food Summit 1996. You can cite both refeences.
  • Line 50: this sentence is not clear: the cited study (Bruening et al. 2017) refers to adverse academic outcome, so please rephrase.

Materials and methods:

  • Lines 82-87: too many repetitions of “large, diverse, state-funded”.
  • Line 112: it would be better to refer at least once to the fact that this is the short form.
  • Lines 125-126: please specify if this is your choice for the distribution of the raw score of there is a specific reference on such scheme.

Results:

  • Table 1: Just a curiosity: were there any student underweight (below 18.5?).
  • Section 3.4: using a logistic regression presume that the response (dependent) variable is binary. It would be better to clarify which is your dependent variable and how you obtain it.

Discussion:

  • Lines 264-266: in Nikolaus et al (2019), the reported figure for the rate of food insecurity is 19.9% (cfr. Fig 1. Prevalence rates of food insecurity among undergraduate college students by assessment protocol, you actually cite and explain these figures very extensively afterwards). Therefore, according to your results, the food insecurity rate is not double but 50% higher.
  • Lines 293-294: again, in this sentence, “2-fold higher” is a bit overestimated, since you classify as food insecure the 34.5% of the student. Your results are very interesting as they are. In my opinion, there is no need to push them further.

Author Response

Please see attached Word doc.

Reviewer 2 Report

The overall report is good enough. Likewise, research is interesting and relevant too. However, here are few observation authors need to address.

  • The methods of calculating weight is not clear, it need to be presented clearly so that reader can understand easily.
  • Authors did not presented how they check the validity of the response. How confidently are the authors that who passed the first screening are food secure? 
  • Overall response rate is just 4.4%, it would be good if the authors explain why such low response occur? or is it normal? any supporting evidence? 
  • For logit regression, OR was calculated but what about the marginal effect?  

Author Response

Please see attached Word Doc.
